# Relationship Between Sigmoid Volvulus Subtypes, Clinical Course, and Imaging Findings

**DOI:** 10.3390/diagnostics15060784

**Published:** 2025-03-20

**Authors:** Kemal Bugra Memis, Sonay Aydin

**Affiliations:** Department of Radiology, Faculty of Medicine, Erzincan Binali Yıldırım University, Basbaglar, 1429th Street, Erzincan 24100, Turkey; sonay.aydin@erzincan.edu.tr

**Keywords:** mesentero-axial sigmoid volvulus, organo-axial sigmoid volvulus, sigmoid volvulus follow-up, imaging in sigmoid volvulus

## Abstract

**Background:** Recent studies indicate that the organo-axial subtype of a sigmoid volvulus is more prevalent than the conventional mesentero-axial subtype. Our study aimed to assess the clinical and radiological findings that differentiate between these two subtypes, as well as to ascertain treatment outcomes and prognostic characteristics. **Methods:** A retrospective review included 54 patients, during which abdominal plain radiographs and computed tomography images were analyzed by two radiologists, and data on recurrence, mortality, and treatment outcomes were documented. **Results:** The mesentero-axial subtype comprised 40 cases (74%). No distinct radiographic findings were observed to differentiate between the two groups. In computed tomography, the sole significant parameter for differentiation was the number of transition zones. The diameter of the segment exhibiting a volvulus was greater in instances of the mesentero-axial subtype. The endoscopic detorsion treatment proved ineffective in five patients within the mesentero-axial sigmoid volvulus cohort. **Conclusions:** Identifying these two types of SV on CT images is essential because of their distinct prognoses and therapeutic results.

## 1. Introduction

A colonic volvulus stands as the third most common reason for large bowel obstruction globally. Colonic volvulus is predominantly found in the sigmoid colon, accounting for 60% to 75% of all cases. It is less frequently observed in the cecum (25% to 40% of cases) and rarely occurs in the transverse colon (1% to 4% of cases) and splenic flexure (1%) [1,2,3].

A sigmoid volvulus (SV) is a frequent underlying factor for acute abdominal pain in older patients. An SV involves the rotation of the sigmoid colon either around the mesocolon or on itself. Due to the obstruction in the colon, the luminal pressure of the intestine gradually increases and causes vascular stasis over time. Progressively, the blood supply to the intestines becomes compromised, resulting in necrosis and, in advanced stages, the development of perforation in the intestinal wall [4,5].

The prevalence of SVs exhibits significant regional disparities due to variations in cultural and dietary habits within the population, differences in altitude, and the potential impact of endemic pathogens. The regions with the greatest incidence include Turkey, Eastern Europe, South America, Russia, the Middle East, Africa, Pakistan, and India. 

Conversely, North America, Western Europe, and Australia are notable for their low incidence rates [4,5,6,7,8]. In countries with a high incidence of SV, it is responsible for 25–50% of cases of colonic obstruction. The annual incidence of SV is 1–2 per 100,000 individuals [9,10].

SVs are more prevalent in individuals over the age of 70 and are more frequently observed in males [3]. The rising prevalence of SVs with advancing age has been ascribed to the elongation of the colon. The primary risk factors for an SV are the presence of a redundant sigmoid colon and a narrow mesentery [11]. Aside from advanced age, being male, and having a redundant sigmoid colon, constipation, neuropsychiatric disorders, postoperative adhesions, and, according to recent studies, pneumoperitoneum in individuals with a history of laparoscopy have been identified as risk factors for SVs [11,12,13,14].

Abdominal radiographs can diagnose an SV in 57–90% of patients [15,16,17]. Plain radiographs have identified significant radiological signs for diagnosing SVs. In the most widely recognized example, referred to as the coffee bean sign, the central portion of the bean is formed by the inner walls of the enlarged colonic loop, while the outer contours of the bean are formed by the outer walls of the dilated colon. This sign is observed in approximately 60% of cases and is one of the signs that has a sensitivity of 100% on abdominal radiographs [15,16,17,18,19]. Other signs with a sensitivity of 100% on abdominal X-rays include the presence of the sigmoid loop apex beneath the left hemidiaphragm, the left-flank overlap sign, and the observation of the sigmoid colon extending further cranially than the transverse colon (referred to as the northern exposure sign) [8,16]. The northern exposure sign, which was initially described by Javors et al. [16], is the most specific sign of an SV and was observed in 87% of cases. In addition to these findings, the absence of gas shadow in the rectum and the presence of significant gas distension in the proximal colon segments can be observed [19].

Abdominal computed tomography (CT) is gaining significance and being utilized more frequently in the differential diagnosis of acute abdominal pain. The diagnostic precision of CT imaging in identifying an SV with acute abdomen etiology is almost 100%. Levsky et al. found that the most accurate indicators on CT scans for diagnosing an SV were the presence of a transition zone in the sigmoid colon (95%) and an abnormal expansion of the sigmoid colon (86%) [18]. In addition to these findings, CT images in patients with an SV have also shown the presence of the whirlpool sign, X-marks-the-spot sign, and split-wall sign [18,20,21]. In complicated cases with delayed diagnosis, the presence of ischemia can be determined by observing pneumatosis intestinalis using the lung window of CT images [22]. According to the literature, it is advised to also examine the CT scanogram. The most frequently observed findings include the lack of gas in the rectum and the presence of the U sign [18,23].

An SV has two primary mechanisms, which are determined by its anatomical axis of rotation: mesentero-axial and organo-axial types (Figure 1). In the mesentero-axial variant of SV, the colon undergoes rotation around the mesenteric axis, resulting in a closed-loop obstruction where the two sides of the colonic loops are in close proximity to each other. A whirlpool sign has been detected on the CT scan at this junction. The ahaustral segment, characterized by a closed-loop obstruction and two transition zones, is situated in the anterior abdominal area. The organo-axial SV type refers to a condition where the colon rotates along its own axis. This type is characterized by a single transition zone and does not result in closed-loop type obstruction. Colon segments proximal to the transition zone are enlarged. The whirlpool sign is also observed in this type of CT scan. According to the literature, the mesentero-axial type is more prevalent. An organo-axial SV has been frequently discussed in the past literature as a distinct entity that is observed infrequently. Nevertheless, it is reported in current literature that this type of SV is also a usual one. Given that a previous occurrence of a volvulus is a significant risk factor for the development of SV, it is crucial to identify which subtype has a higher probability of recurring during patient follow-up. In addition, there are incredibly few studies in the literature that compare these two subtypes in terms of clinical-radiological findings, treatment success, recurrence, and mortality [11,14,24].

The goal of the research was to ascertain the occurrence of two types of SV and to assess the associations between these types in relation to clinic-radiological findings, effectiveness of treatment, recurrence rates, and mortality.

## 2. Materials and Methods

### 2.1. Ethical Aspects

This study obtained ethical approval from the Clinical Research Ethics Committee at Erzincan Binali Yıldırım University (Protocol number: EBYU-KAEK-2024-11/01/Date: 22 August 2024). Each patient gave written informed consent for the publication of this article using their data.

### 2.2. Study Design

Imaging reports of all contrast-enhanced abdominal CT examinations performed between January 2016 and January 2024 in the Radiology Department of our hospital were scanned using the search term “volvulus”. Out of the patients who were scanned, a total of 86 cases of sigmoid volvulus were found. Our study will include patients with a surgically confirmed sigmoid volvulus who underwent a contrast-enhanced abdominal CT examination at our hospital within 24 h of symptom onset and whose follow-up and treatment occurred at our facility between January 2016 and January 2024. Individuals whose clinical and radiologic findings indicated the presence of an SV but a volvulus was not observed during the operation, as well as individuals whose necessary records could not be accessed due to their ongoing treatment in another healthcare facility were subsequently excluded from the study. Out of 86 patients retrospectively screened, 8 were excluded due to the absence of an SV during surgery, and 24 were excluded because their treatment and follow-up occurred at another hospital, rendering data access impossible. A total of 54 patients were included in this study. Figure 2 illustrates a flow chart depicting the study design.

### 2.3. CT Protocol

Contrast-enhanced abdominal CT images were obtained using a 128-slice CT scanner (Somatom go.Top, Siemens Healthcare, Forchheim, Germany). The standard acquisition protocol was used to maintain a dose of 80 kVp in all patients. The device automatically adjusted the mAs (60–220) based on the patient’s weight. The thickness of the slice was 1 mm. The CT examination covered the region extending 2 cm above the diaphragm to the lesser trochanter in the craniocaudal direction. The CT scans were conducted during the portal venous phase using an iohexol (80–100 mL), which was injected through the antecubital vein at a rate of 3 mL/s using an automatic injector. Afterwards, a 40 mL bolus of saline was administered.

A PACS system (Akgun PACS Viewer v7.5; Akgun Software, Ankara, Turkey) was used to analyze the cross-sectional images. Thin-section axial images were evaluated at the workstation and reformats were obtained from coronal and sagittal planes.

### 2.4. Evaluation of Radiological Images

Two experienced radiologists with 11 and 15 years of experience retrospectively analyzed contrast-enhanced abdominal CT images and plain abdominal radiographs, and the type of volvulus was determined by reaching a consensus based on the course of the bowel and mesentery. This study assessed various factors on plain abdominal radiographs, including the presence of an inverted ‘U’ shape appearance and coffee bean sign in the sigmoid, dilatation in the proximal colon segments, absence of rectal gas, the left-flank overlap sign, the northern exposure sign, and the presence of the sigmoid loop apex beneath the left hemidiaphragm. For each case, the redundant sigmoid colon, the number of transition zones in the colon, the maximum diameter of the enlarged sigmoid colon, and the presence of specific signs such as the whirlpool sign, X-marks-the-spot sign, and split-wall sign were documented in the contrast-enhanced abdominal CT images. Furthermore, if the patient exhibited indications of intestinal ischemia, such as simultaneous reduction in contrast enhancement of the intestinal wall and the presence of pneumatosis, these observations were documented.

### 2.5. Evaluation of Clinical Findings and Treatment Results

White blood cell (WBC) count, neutrophil count, C-reactive protein (CRP), and lactate levels were recorded from the laboratory results. The presence of comorbid diseases such as diabetes mellitus, history of chronic constipation, psychiatric diseases, and neurological diseases were also investigated. Patients who had a medical history of SV were identified. Medical, surgical, or endoscopic interventions were administered to each patient following diagnosis, and the outcomes were documented. Furthermore, mortality rates within the initial 30 days post-hospital admission were assessed.

### 2.6. Statistical Analysis

Data were analyzed using the Package for Social Sciences (SPSS) 25 for Windows (IBM SPSS Inc., Chicago, IL, USA). Normal distribution of the data was evaluated using the Kolmogorov–Smirnov test. Numerical variables with normal distribution are shown as mean  ±  standard deviation. The variables not with normal distribution are shown as minimum–maximum values. Categorical variables are shown as numbers and percentages. For the comparison of numerical variables between two sigmoid volvulus groups, the Mann–Whitney U test and Student’s *t*-test were used. For the comparison of categorical variables, the Fisher’s exact or Chi-squared tests were applied.

A two-tailed value of *p*  <  0.05 was considered statistically significant.

## 3. Results

A total of 54 patients, 12 (22%) female and 42 (78%) male, were included in this study. The median age was 66 (range 44–87).

CT images were initially assessed, revealing that 14 patients (26%) presented with organo-axial SV and 40 patients (74%) presented with mesentero-axial SV.

The demographic, clinical, and radiological characteristics of the patients are explained in Table 1.

The predominant complaints at initial presentation were abdominal distension in 52 (96%) patients and cramping abdominal pain in 50 (93%) patients. The predominant risk factors or comorbidities included chronic constipation in 35 (65%) patients and diabetes mellitus in 17 (31%) patients.

The plain radiographic findings and general radiological signs of SV are presented in Table 2. Classic radiographic signs for SV were identified in 43 out of 54 cases (80%). Five patients (9%) solely exhibited ileus findings, whereas the remaining six (11%) presented no pathological findings. The predominant findings on abdominal radiographs included an inverted-U-shaped distended sigmoid loop in 42 patients (78%), absence of gas in the rectum with 40 patients (74%), and the coffee bean sign in 37 patients (69%). Twenty patients (37%) exhibited the northern exposure sign. The least common findings were the left-flank overlap sign and the presence of the sigmoid loop apex beneath the left hemidiaphragm, observed in three patients (5.5%) each.

Table 3 shows the findings from contrast-enhanced abdominal CT imaging in patients with SV. The predominant CT finding was the detection of one or more transitional zones in the sigmoid colon in 52 patients (96%). Subsequently, sigmoid colon overdistension was observed in 48 patients (89%), and a whirlpool sign was identified in 40 patients (74%). The X-marks-the-spot sign, indicative of SV, was identified in 29 patients (46%), while the split-wall sign was observed in 21 patients (39%).

Abdominal plain radiographic findings, including specific signs, were determined to have no significant impact on differentiating between mesentero-axial and organo-axial SV subtypes. Our abdominal CT findings and measurements indicated significant parameters that may aid in discriminating between mesentero-axial and organo-axial subtypes. The most notable finding was the number of transition zones identified within the colonic segments. In mesentero-axial type SV, 36 (90%) of 40 patients exhibited two transition zones, while 4 (10%) presented with a single transition zone. In the organo-axial SV type, the transition zone could not be identified in 2 out of 14 patients. Among the remaining 12 patients, 11 (92%) exhibited a single transition zone, while 1 (8%) patient presented with two transition zones. No notable selectivity was detected in additional parameters and indicators assessed in CT for the differentiation of these two subtypes.

The mean maximum diameter of dilated sigmoid colon segments in the study population was 7.89 ± 1.46 cm (range: 6.05–11.17 cm). The mean maximum diameter of the sigmoid colon was 8.06 ± 1.78 cm (range: 6.15–11.17 cm) in the mesentero-axial group and 7.41 ± 1.26 cm (range: 6.05–9.81 cm) in the organo-axial group. The maximum sigmoid colon diameter did not provide a statistically significant distinction between the two SV subgroups (*p* = 0.60).

CT images for four of our patients showed findings indicative of ischemia and necrosis, including intestinal wall thickening, decreased mural contrast enhancement, and pneumatosis intestinalis. Also, detorsion endoscopy identified intestinal necrosis in three additional patients. A total of seven patients (13%) suffered intestinal ischemia caused by SV. Among these, four were classified as mesentero-axial, while the remaining three were categorized within the organo-axial SV group. The development of ischemia did not differ significantly between the two groups (*p* = 0.11).

Upon analyzing the initial treatment, we found that 17 patients (31.5%) received endoscopic detorsion, while 37 patients (68.5%) underwent surgical intervention, which usually included colectomy procedures. Volvulus had been successfully resolved in 9 out of 17 patients (53%) who underwent detorsion. Three patients (18%) showing necrosis were referred for surgical colectomy. In the remaining five patients (29%), endoscopic detorsion proved ineffective, necessitating surgical referral. All of these five patients had mesentero-axial SV.

In the initial 30 days, SV-related mortality was observed in 12 patients (22.2%). Intestinal necrosis was identified in five of these patients. Endoscopic detorsion treatment was primarily administered to the remaining five patients. Of the deceased patients, seven exhibited mesentero-axial splenic volvulus, while the remaining five presented with organo-axial splenic volvulus. The disease-related mortality rate was 17.5% for mesentero-axial SV and 35.7% for organo-axial SV in our study (*p* < 0.05).

Figure 3 and Figure 4 illustrate plain radiographs and CT images of patients with mesentero-axial and organo-axial sigmoid volvulus, respectively.

## 4. Discussion

Our study examined the distinct radiological findings of mesentero-axial and organo-axial SV subtypes as defined by Bernard et al. in 2010 [25]. We determined that the presence of multiple transition zones signifies mesentero-axial SV, while a single transition zone indicates organo-axial SV. Furthermore, we discovered that organo-axial SV necessitates more frequent surgical intervention and exhibits a greater incidence of recurrence; this underscores the significance of differentiating these two subtypes in the selection of treatment for patients and the recurrence of an SV.

Recent studies indicate that organo-axial SV occurs with greater frequency or is comparable in prevalence to mesentero-axial type [24,25,26]. Toh et al. reported that the organo-axial subtype had been discovered in 68% of patients with SV [11]. In another study conducted by Moloney et al., organo-axial and mesentero-axial subtypes were documented to have nearly equal prevalence in SV patients [24]. In contrast to these articles, 76% of our study population exhibited mesentero-axial SV, indicating that the mesentero-axial subtype was three times more prevalent than organo-axial SV in our cohort. This rate closely aligns with the findings presented in the WSES guidelines by Tian et al., indicating that the mesentero-axial subtype constitutes 75% of SV cases, whereas the organo-axial subtype accounts for the remaining 25% [14]. The varying results indicate that the prevalent subtype of SV may differ across populations. In areas where SVs are endemic, like Turkey, the mesentero-axial subtype may predominate; in contrast, in Western nations where SVs are less prevalent, the organo-axial subtype may be more frequent. To substantiate this argument, organo-axial type SV was almost never documented in the largest single-center SV case series globally, which included over a thousand patients, as reported by Atamanalp et al. It was reported that nearly all cases in this largest series were of the mesentero-axial type SV [8,27]. Although it is mentioned in the literature that organo-axial SV is an early form of mesentero-axial SV, there is no consensus on this issue.

An SV is the third most prevalent cause of colonic ileus [28]. Conventional abdominal radiographs are the most commonly ordered scans for patients presenting to the emergency department with ileus symptoms, including acute abdominal pain, distension, and prolonged constipation. Typical findings for an SV can be identified in up to 90% of patients on the radiographs [29]. Nonetheless, certain studies indicate that the efficacy of radiographs in diagnosing SV is approximately 65% [15]. In our patient cohort, diagnostic findings for SV were found on radiographs at a rate of 80%, which correlates with the average rates documented in these two studies. Moreover, there are no radiographic signs available to distinguish the subtypes of SV in our study or in the current literature.

When clinical findings indicate SV, abdominal CT scans may be conducted in patients to improve diagnostic accuracy with respect to radiographs. Moreover, CT imaging plays a crucial role in ruling out aetiologies such as malignancies and diverticulitis, which are prevalent causes of colonic type ileus. The mesentero-axial and organo-axial subtypes can be readily distinguished on abdominal CT images. Our study demonstrates that the most significant finding in this distinction is the number of transition zones. We have demonstrated that cases with a single transition zone are classified as organo-axial, whereas cases with two transition zones, specifically closed-loop type stenosis, typically fall under mesentero-axial SV. This outcome parallels what was documented in the literature by Toh et al. [11]. Additionally, the literature indicates that the split-wall sign, noted in the early phases of bowel rotation, is a more prevalent CT finding in patients with organo-axial SV. Nonetheless, in our findings, while this observation was more prevalent in the organo-axial type, it did not attain statistical significance [11,18,20,30]. Another CT finding referenced in the literature to distinguish these two subtypes is the X-marks-the-spot sign. This sign manifests in the advanced stages of intestinal rotation, arising from the adjacent transition zones in the mesentero-axial SV, and is consequently more prevalent in this type. In our findings, this sign was comparatively more prevalent in the mesentero-axial type; however, the results did not attain statistical significance [11,18,30].

The most recent research conducted by Moloney et al. [24] on mesentero-axial and organo-axial SV involving 117 patients evaluated the direction of rotation in the axial and coronal planes. The authors have demonstrated a significant correlation between clockwise rotation in one plane and anticlockwise rotation in the opposing plane. They also emphasized that these rotational directions are significant findings in identifying the type of volvulus. It has been shown that the mesentero-axial SV subtype is linked to anti-clockwise rotation in the axial plane and clockwise rotation in the coronal plane, while the organo-axial SV subtype is associated with clockwise rotation in the axial plane and anti-clockwise rotation in the coronal plane [24]. The determination of these rotation directions, first established in the literature, may serve as a reliable alternative to the number of transition zones, which is the most definitive indicator in distinguishing organo-axial and mesentero-axial SV subtypes, as demonstrated in our study.

Differentiating between these two types of SV on CT images is crucial due to their significantly different prognoses and treatment responses. Given the closed-loop obstruction in the mesentero-axial SV, it is unsurprising that intestinal ischemia and necrosis may progress more rapidly. Therefore, the incidence of severe complications and mortality rates are higher in this subtype. Moreover, a number of studies indicate a significant failure rate of endoscopic detorsion in mesentero-axial SV conditions. In the majority of cases, concomitant ischemia in the intestine is regarded as the primary cause [30,31]. The study by Atamanalp et al., conducted in our geographical region and representing the largest single-center experience globally with over one thousand SV patients, demonstrated that the treatment success rates of non-operative and operative detorsions were comparable in patients without intestinal ischemia [27]. In our study population, surgical methods are utilized more often than endoscopic detorsion for treating SV patients despite the literature usually favoring endoscopic procedures. The explanation for this condition appears to be that the selected treatment strategy may differ based on the clinicians’ experience and practices at the health centers. Atamanalp et al. [27] and Moloney et al. [24] demonstrated that endoscopic detorsion techniques yielded lower mortality and morbidity rates. At this point, it is crucial to rule out ischemia on CT scans to facilitate the referral of the patient for endoscopic detorsion. Conversely, the presence of ischemic findings necessitates surgical intervention, which should be conducted promptly to mitigate mortality risks. In our study, all five patients without bowel ischemia, in whom endoscopic detorsion was ineffective as the primary treatment, belonged to the mesentero-axial SV group. While that result aligns with the existing literature, the non-randomized selection of initial treatment (endoscopic detorsion versus laparotomy) in our study precludes definitive conclusions regarding treatment choice and efficacy based on subtypes.

Our analysis revealed an overall mortality rate related to SV of 22.2%. Atamanalp et al. [27] reported a mortality rate of under 1% in uncomplicated non-ischemic SV patients within the largest cohort of SV patients investigated. In addition, the mortality rate in SV patients with complicated intestinal ischemia was observed to be 16.5%. The main reason for the higher SV-related mortality rate, in contrast to most literature data, is our failure to categorize complicated patients with intestinal ischemia into a different group. The study that separates these complicated patients would reveal a reduced mortality rate in the cohort without associated intestinal ischemia. Nearly fifty percent of patients having SV-related mortality also exhibited simultaneous intestinal ischemia. Another reason for our higher mortality rate related to SV is the limited number of individuals with the organo-axial SV subtype. The literature indicates that the mortality rate in comparable studies with a limited sample size can exceed our findings. Toh et al. [11] reported a mortality rate of 31% due to SV in a study including 19 patients.

When the largest diameter was measured at the level of the sigmoid loop with volvulus, it was found that the median diameter was higher in the mesentero-axial SV group than in the organo-axial SV group. Moloney et al. recently reported that an increase in the maximum diameter was closely linked to SV recurrence [24]. Thus, it can be predicted that mesentero-axial SV cases exhibiting a bigger increase in sigmoid loop diameter may experience a higher recurrence rate.

Given the need for a thorough evaluation of this subject, our research should have included a more extensive study cohort. However, given the low prevalence of SV, at 1–2 per 100,000, and our stringent inclusion criteria, the total number of eligible patients for this research was limited. This represents the primary limitation of our study. We would like to underscore that our analysis encompasses the second biggest patient group (Table 4) relative to comparable studies in the recent literature [11,24,25].

Our research exhibited certain limitations. The retrospective design constitutes a significant limitation. Acquiring clinical data from file records may lead to selection bias. Another one is our non-randomized methodology for initial treatment selection. Consequently, the selection of treatment based on subtypes and the assessment of treatment efficacy lacked objectivity. A further limitation is that, despite the patient population in our study being comparable to recent studies, the organo-axial SV group had a notably low number of patients. This situation prevents certain results from obtaining statistical significance and affects the generalisability of findings related to the organo-axial subtype. Additionally, nearly fifty percent of the patients in the study cohort had to be excluded due to their continuation of follow-up and treatment at a different hospital, resulting in a lack of treatment and prognosis data.

## 5. Conclusions

In summary, our research indicates that the mesentero-axial variant of sigmoid volvulus is more prevalent than the organo-axial variant in our region, which is recognized as endemic for sigmoid volvulus. According to our results, CT can be a useful tool to differentiate between the organo-axial and mesentero-axial subtypes.

## Figures and Tables

**Figure 1 diagnostics-15-00784-f001:**
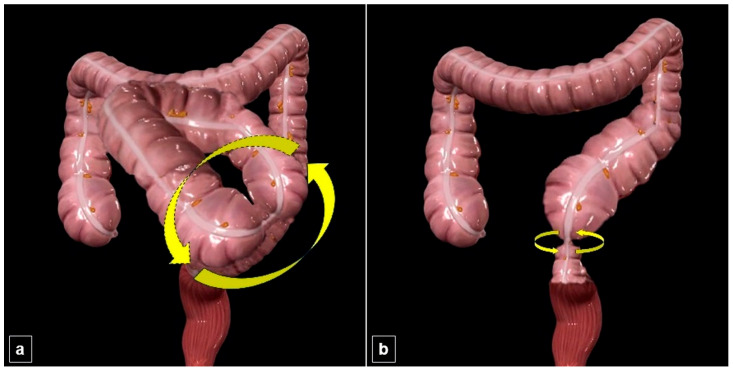
Two mechanisms of sigmoid volvulus. (**a**) The three-dimensional image of colon segments illustrates the mechanism of mesentero-axial sigmoid volvulus. Two transition zones are identified in the proximal and distal parts of the sigmoid colon segment exhibiting a closed-loop obstruction. (**b**) An image showing the mechanism of organo-axial sigmoid volvulus reveals only one transition zone.

**Figure 2 diagnostics-15-00784-f002:**
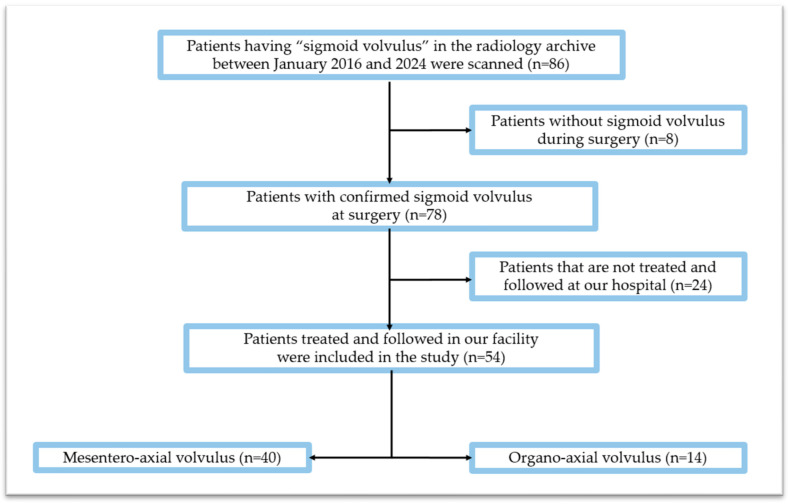
Diagram showing the study population.

**Figure 3 diagnostics-15-00784-f003:**
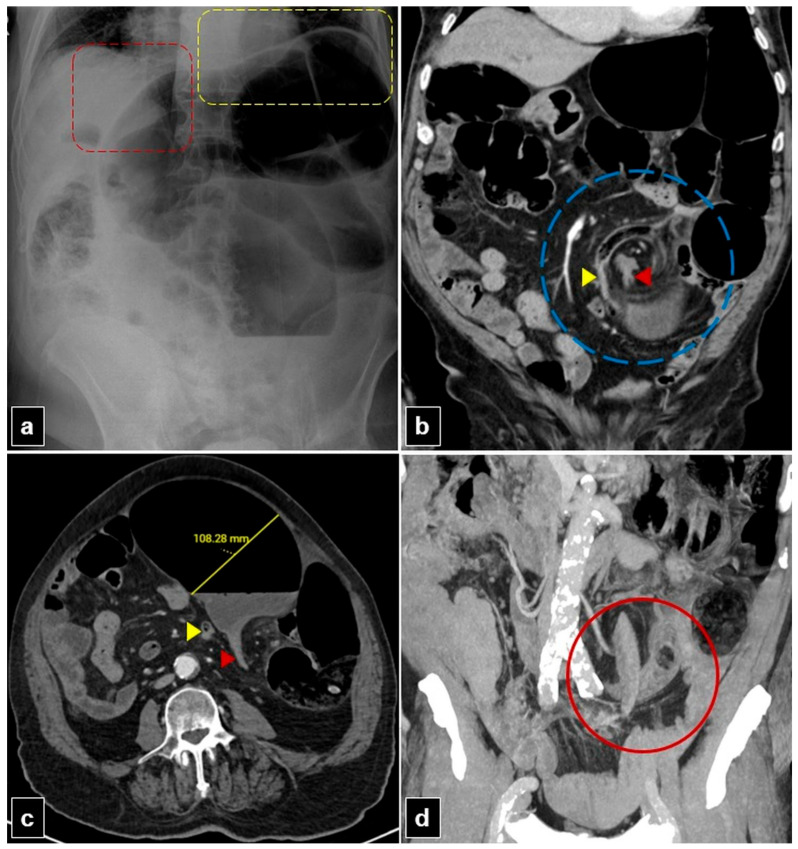
An 80-year-old male patient with a mesentero-axial sigmoid volvulus presented with abdominal pain and constipation for seven days. (**a**) The plain abdomen radiograph reveals significant distension of the sigmoid loop, characterized by an inverted “U” shape, absence of rectal air lucency, the northern exposure sign, liver overlap sign (red dashed quadrilateral), and the sigmoid apex located beneath the left hemidiaphragm sign (yellow dashed quadrilateral). (**b**) The coronal plane abdominal computed tomography (CT) image reveals the mesenteric whirlpool sign (blue dashed circle) and transition zones located proximally (yellow arrowhead) and distally (red arrowhead) to the rotating sigmoid loop. (**c**) The axial plan abdominal CT image shows the degree of distension in the sigmoid colon and the transition zones situated proximally (yellow arrowhead) and distally (red arrowhead) to the rotating sigmoid loop. (**d**) The coronal plane maximum intensity projection (MIP) CT image of the abdomen shows an x-marks-the-spot sign (red circle) created by the proximal and distal transition zones.

**Figure 4 diagnostics-15-00784-f004:**
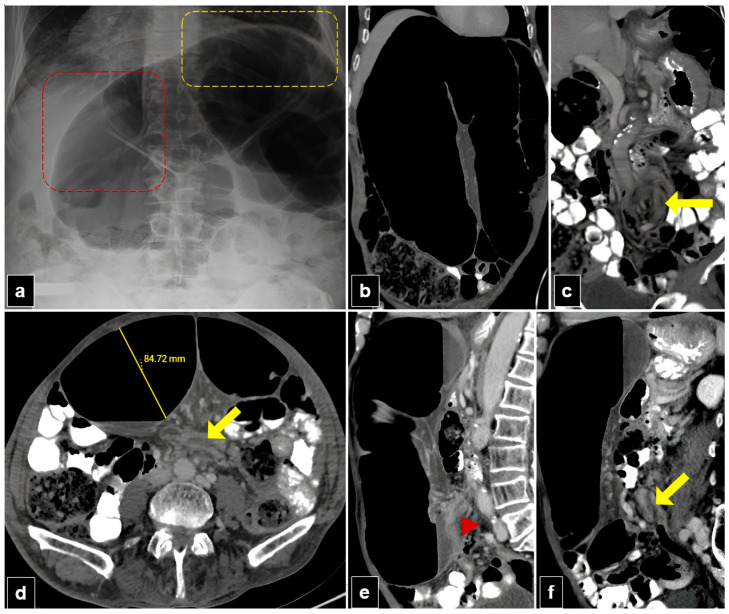
A 77-year-old male patient with organo-axial sigmoid volvulus presented with abdominal pain and distension for two days. (**a**) The plain abdomen radiograph reveals significant distension of the sigmoid loop, characterized by an inverted “U” shape, liver overlap sign (red dashed quadrilateral), and the sigmoid apex located beneath the left hemidiaphragm sign (yellow dashed quadrilateral). (**b**,**c**) Abdominal computed tomography (CT) images in the coronal plane show the inverted U-appearance of the distended sigmoid loop, and only one transition zone is present. Note the split wall sign caused by mesenteric fat tissue at the level of the transitional zone (yellow arrow). (**d**) The axial plan abdominal CT image shows the degree of distension in the sigmoid colon and the split wall sign located in the transition zone (yellow arrow). (**e**,**f**) The sagittal plane CT images of the abdomen show a single transition zone (red arrowhead) and a split wall sign (yellow arrow).

**Table 1 diagnostics-15-00784-t001:** The relationship between the demographic and clinical features of the patients.

	Mesentero-Axial SV (*n* = 40)	Organo-Axial SV (*n* = 14)	*p*-Value
Sex *n* (%)			
Female	8 (20)	4 (28.5)	0.150
Male	32 (80)	10 (71.5)	
Age (years) M (min–max) ^1^	68 (52–87)	64 (44–82)	0.732
Laboratory Parameters M (min–max) ^1^			
WBC (×10^3^/µL)	6.69 (5.2–12.21)	6.1 (4.79–8.87)	0.268
Neutrophil (×10^3^/µL)	4.72 (3.26–9.8)	4.25 (3.01–7.12)	0.245
CRP (mg/L)	56.5 (24.5–98.5)	53.25 (23.5–107)	0.546
Lactate level (mmol/L)	497.5 (219–665)	392 (239–704)	0.325
Complaints of Admission *n* (%)			
Crampy abdominal pain	39 (97.5)	11 (78.5)	0.105
Abdominal distension	39 (97.5)	13 (92.8)	0.420
Constipation	24 (60)	8 (57.1)	0.721
Vomiting	9 (22.5)	3 (21.4)	0.855
Risk factors *n* (%)			
Chronic constipation	27 (67.5)	8 (57.1)	0.341
Neuropsychiatric disorders	12 (30)	4 (28.5)	0.820
Diabetes mellitus	12 (30)	5 (35.7)	0.447
Prior abdominal laparoscopy	10 (25)	3 (21.4)	0.465
History of previous SV ^2^	11 (27.5)	5 (35.7)	0.195

^1^ M: median, min: minimum, max: maximum. ^2^ SV: sigmoid volvulus.

**Table 2 diagnostics-15-00784-t002:** Plain abdominal radiograph findings in two subtypes of sigmoid volvulus.

	Mesentero-Axial SV (*n* = 40)	Organo-Axial SV (*n* = 14)	
	*n* (%)	*n* (%)	*p*-Value
Classic radiographic signs of SV	32 (80)	11 (78.6)	1.000
Isolated colonic ileus findings	4 (10)	1 (7.2)	1.000
No pathology found	4 (10)	2 (14.2)	1.000
Inverted-U-shaped distended sigmoid	31 (77.5)	11 (78.6)	0.850
Absence of gas in the rectum	29 (72.5)	11 (78.6)	0.445
Coffee bean sign	28 (70)	9 (64.2)	0.432
Northern exposure sign	15 (37.5)	5 (35.7)	0.886
Left-flank overlap sign	2 (5)	1 (7.1)	1.000
Sigmoid apex beneath the left hemidiaphragm	2 (5)	1 (7.1)	1.000

**Table 3 diagnostics-15-00784-t003:** Abdominal computed tomography findings in two subtypes of sigmoid volvulus.

	Mesentero-Axial SV (*n* = 40)	Organo-Axial SV (*n* = 14)	
	*n* (%)	*n* (%)	*p*-Value
Redundant sigmoid colon	28 (70)	11 (78.6)	0.455
Transitional zone(s)			
None	1 (2.5)	1 (7.2)	
One	1 (2.5)	13 (92.8)	<0.001
Two	38 (95)	0	<0.001
Sigmoid colon overdistension	36 (90)	12 (85.7)	0.520
Whirlpool sign	30 (75)	10 (71.5)	0.645
Split-wall sign	15 (37.5)	6 (42.9)	0.580
X-marks-the-spot sign	22 (55)	7 (50)	0.499

**Table 4 diagnostics-15-00784-t004:** Recent studies on similar topics and patient populations.

Authors (Year)	Patient Population		
	Mesentero-Axial SV (*n*)	Organo-Axial SV (*n*)	Total (*n*)
Toh et al. (2023) [11]	6	13	19
Moloney et al. (2024) [24]	39	41	80
Bernard et al. (2010) [25]	6	17	23
Our study	40	14	54

## Data Availability

The data supporting the findings of this study are available from the corresponding author (S.A.) upon request.

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
