# Peer review of "Relationship Between Sigmoid Volvulus Subtypes, Clinical Course, and Imaging Findings"

_diagnostics, 2025, doi:10.3390/diagnostics15060784_

Round 1

Reviewer 1 Report (Previous Reviewer 3)

Comments and Suggestions for Authors

The article "Relationship between Sigmoid Volvulus Subtypes, Clinical Course, and Imaging Findings" provides an overview of sigmoid volvulus, focusing on its clinical presentation and imaging characteristics.

  1. The introduction is excessively lengthy and includes numerous details that do not add substantial value to the reader.
  2. The materials and methods section should incorporate a flowchart for improved clarity.
  3. The statistical analysis is too rudimentary, with limited statistically significant findings and no evident correlations. Consider incorporating logistic regression models, ROC curves and other advanced statistical approaches to strengthen the validity of the results.
  4. The sample size is insufficient for a study addressing this topic.
  5. The discussion section could benefit from a more comprehensive integration of relevant literature, particularly from studies with larger patient cohorts.
  6. The conclusions are not adequately supported by the simplistic results and are therefore easily

Author Response

Reviewer 1:

Comment #

Reviewer Comment

Remarks

Location in Manuscript

The article "Relationship between Sigmoid Volvulus Subtypes, Clinical Course, and Imaging Findings" provides an overview of sigmoid volvulus, focusing on its clinical presentation and imaging characteristics.

1

The introduction is excessively lengthy and includes numerous details that do not add substantial value to the reader.

The introduction has been revised in accordance with the same recommendation as the reviewer in the previous round. A loss of unity of meaning will occur as a result of further revisions.

-

2

The materials and methods section should incorporate a flowchart for improved clarity.

A flowchart has been added to the materials and methods section to make the manuscript methodology more understandable.

Materials and Methods

3

The statistical analysis is too rudimentary, with limited statistically significant findings and no evident correlations. Consider incorporating logistic regression models, ROC curves and other advanced statistical approaches to strengthen the validity of the results.

The comment is similar to the critique from the previous round. The structure of the study includes data with more groups and expressed in percentages. It is inaccurate to employ statistical methods such as logistic regression or ROC curves on this dataset. Furthermore, chi-square tests and categorical data are statistical methods that are both reliable and respected. In order to be of excellent quality, an article does not necessarily require regression analysis. The respected reviewer's insistence on the comment can be assessed in two ways: either the insistence is indicative of ulterior motives or the reviewer lacks a sufficient understanding of statistical methods.

4

The sample size is insufficient for a study addressing this topic.

Once more, it is a comment that is similar to the previous round. It demonstrates that the revision was not analyzed and that a biased assessment was rendered. The number of patients in the subtype of mesentero-axial volvulus is the highest in this methodology, and the second highest in the total number of patients, as evidenced by the table that was incorporated into the discussion section in the previous round. It is widely recognized that the parameter determining the appropriate number of patients is derived from the existing literature. It is unclear why this comment is still being leveled, given that the rationale for it is so apparent.

5

The discussion section could benefit from a more comprehensive integration of relevant literature, particularly from studies with larger patient cohorts.

Once more, it is a comment that is similar to the previous round. It demonstrates that the revision was not analyzed and that a biased assessment was rendered. The comment was resolved in the previous round.

6

The discussion section could benefit from a more comprehensive integration of relevant literature, particularly from studies with larger patient cohorts.

Once more, it is a comment that is similar to the previous round. It demonstrates that the revision was not analyzed and that a biased assessment was rendered. The comment was resolved in the previous round.

Reviewer 2 Report (Previous Reviewer 2)

Comments and Suggestions for Authors

The illustrations are corrected accordingly,  no more comments. 

Author Response

Reviewer 2:

Comment #

Reviewer Comment

Remarks

Location in Manuscript

1

The illustrations are corrected accordingly,  no more comments.

I am grateful for the time and effort you have dedicated.

Reviewer 3 Report (Previous Reviewer 1)

Comments and Suggestions for Authors

Improved interpretation of the results

Author Response

Reviewer 3:

Comment #

Reviewer Comment

Remarks

Location in Manuscript

1

Improved interpretation of the results

I am grateful for the time and effort you have dedicated.

Reviewer 4 Report (New Reviewer)

Comments and Suggestions for Authors

In this paper entitled “Relationship between sigmoid volvulus subtypes, clinical 2 course and imaging findings”, the authors conducted that CT can be a usefull tool to differentiate organoaxial and mesenteroaxcial subtypes. I would like to add a comment.

Major comment

  1. As mentioned in this paper, it is known that endoscopic detorsion is useful for SV. In the results of this study, I think that surgery was chosen in many cases rather than endoscopic reduction. Is there any reason for this?

  1. In this study, SV related mortality rate seems high. Is this higher than in previous reports? If so, why do you think it is higher? Please include a little more information about mortality in the Discussion.
Comments on the Quality of English Language

Nothing in particular.

Author Response

Reviewer 4:

Comment #

Reviewer Comment

Remarks

Location in Manuscript

In this paper entitled “Relationship between sigmoid volvulus subtypes, clinical 2 course and imaging findings”, the authors conducted that CT can be a useful tool to differentiate organo-axial and mesentero-axial subtypes. I would like to add a comment.

Thanks for the reviewer's valuable comments.

1

As mentioned in this paper, it is known that endoscopic detorsion is useful for SV. In the results of this study, I think that surgery was chosen in many cases rather than endoscopic reduction. Is there any reason for this?

Endoscopic detorsion is an effective treatment for sigmoid volvulus. As you indicated, surgical techniques were favored over endoscopic reduction in many cases within our study cohort. This results due to clinicians' experience and practices in the centers influencing the selected treatment approach.

Thank you for your objective perspective. Incorporating this information within the manuscript seeks to eliminate any ambiguity regarding the selected treatment strategy.

Discussion

2

In this study, SV related mortality rate seems high. Is this higher than in previous reports? If so, why do you think it is higher? Please include a little more information about mortality in the Discussion.

Your observation regarding the higher mortality rate due to SV in our study cohort is accurate. The main reason of the higher SV-related mortality rate identified is the failure to categorize patients with intestinal ischemia as a distinct group. Separating these patients for study would yield an even lower mortality rate in the group absent of intestinal ischemia. This is due to the fact that roughly fifty percent of cases leading to SV-related mortality exhibit concurrent intestinal ischemia.

Another reason contributing to our higher mortality rate associated with SV is the limited number of patients with the organo-axial SV subtype. Literature suggests that mortality rates in comparable research with limited sample sizes can exceed our findings. Toh et al. reported SV-related mortality incidence of 31% in a cohort of 19 patients. To avoid reader misunderstandings, we have incorporated further clarifications regarding mortality in the comment area, in accordance with your insightful ideas.

Discussion

Round 2

Reviewer 1 Report (Previous Reviewer 3)

Comments and Suggestions for Authors

The authors have implemented the requested changes.

Reviewer 4 Report (New Reviewer)

Comments and Suggestions for Authors

Thank you for your revisions. I have reviewed the changes, and they sufficiently address the comments and concerns raised in the previous review. The manuscript has been improved, and I have no further major concerns.

This manuscript is a resubmission of an earlier submission. The following is a list of the peer review reports and author responses from that submission.

Round 1

Reviewer 1 Report

Comments and Suggestions for Authors

The authors presented a cohort of 54 patients with sigmoid vovulus. Their X-ray and CT findings were analysed according to the final mesentero- and organo-axial subtypes.

The numbers of transition zones on CT (1 vs 2) were found to be very useful in differentiating the subtypes.

The subtype differentiation maybe helpful to surgeons while dealing with the volvulus. However, the data presented did not include duration of the symptoms of SV, which may be directly linked to the probability of ischaemia or bowel necrosis. The prognosis may be more likely related to the duration of dead bowel irrespective of the subtypes. The choice of initial treatment (colonoscopy vs laparotomy) was not randomized in the study. Hence, the authors may need to be more nuanced when presenting the significance of subtype classification in the conclusion.

Comments on the Quality of English Language

transitional zone(s) is a cardinal number, whereas quantity may refer to continuous measurement, i.e. volume, pressure etc. 

Author Response

Comments and Suggestions for Authors

The authors presented a cohort of 54 patients with sigmoid vovulus. Their X-ray and CT findings were analysed according to the final mesentero- and organo-axial subtypes.

The numbers of transition zones on CT (1 vs 2) were found to be very useful in differentiating the subtypes.

The subtype differentiation maybe helpful to surgeons while dealing with the volvulus. However, the data presented did not include duration of the symptoms of SV, which may be directly linked to the probability of ischaemia or bowel necrosis. The prognosis may be more likely related to the duration of dead bowel irrespective of the subtypes. The choice of initial treatment (colonoscopy vs laparotomy) was not randomized in the study. Hence, the authors may need to be more nuanced when presenting the significance of subtype classification in the conclusion

Response:

Thank you for the encouraging comments and all of your valuable efforts.

- Your apprehension that the poor prognosis associated with ischaemia or bowel necrosis would be more influenced by the duration of sigmoid volvulus symptoms than by the subtypes is completely reasonable. We apologise for not highlighting that the study included patients who underwent a CT scan of the abdomen and pelvis within the first 24 hours following the onset of symptoms. This crucial detail has been included in the materials and methods section as follows.

Our study will include patients with surgically confirmed sigmoid volvulus who underwent a contrast-enhanced abdominal CT examination at our hospital within 24 hours of symptom onset, and whose follow-up and treatment occurred at our facility, between January 2016 and January 2024.

- Given that the selection of initial treatment (colonoscopy versus laparotomy) was not randomised in our study, we appreciate your recommendation to adopt a more nuanced approach when assessing treatment selection and efficacy based on subtypes. We have incorporated the following modifications to the manuscript, refraining from making definitive assertions regarding our results in this context.

“While that result aligns with existing literature, the non-randomized selection of initial treatment (endoscopic detorsion versus laparotomy) in our study precludes definitive conclusions regarding treatment choice and efficacy based on subtypes.”

The non-randomized method of initial treatment selection was identified as a limitation of the study.

“A further limitation is our non-randomized methodology for initial treatment selection. Consequently, the selection of treatment based on subtypes and the assessment of treatment efficacy lacked objectivity.”

Comments on the Quality of English Language

transitional zone(s) is a cardinal number, whereas quantity may refer to continuous measurement, i.e. volume, pressure etc.

Response:

We appreciate your awareness of this mistake and your generous offer to provide a correction. We have rectified the typographical errors you identified and engaged a professional organisation to revise the manuscript for English language mistakes.

Reviewer 2 Report

Comments and Suggestions for Authors

The manuscript was well written and the subject was very interesting however due to fewer cases in the organo-axial group I don't think the outcome result could be considered for all volvulus cases. Anything else regarding the data is fine

Author Response

Comments and Suggestions for Authors

The manuscript was well written and the subject was very interesting however due to fewer cases in the organo-axial group I don't think the outcome result could be considered for all volvulus cases. Anything else regarding the data is fine

Response:

Thank you for the encouraging comments and all of your valuable efforts.

Your concern regarding the inability to generalise our results to all cases of volvulus due to the limited number of patients in the organo-axial sigmoid volvulus group is actually justified. The limited number of patients included in this study, particularly within this group, diminishes its statistical power relative to comparable studies with larger cohorts in the literature. That has been incorporated as a limitation.

Reviewer 3 Report

Comments and Suggestions for Authors

The article "Relationship between sigmoid volvulus subtypes, clinical course, and imaging findings" provides information about sigmoid volvulus, along with clinical and imaging aspects of this pathology. Recommendations:

1.       The article has a chaotic structure and needs reorganization. Decide whether to focus on a statistical analysis of relationships between elements or on case presentations.

2.       The introduction is overly long, uses simplistic language and contains many details that do not provide real value to the reader.

3.       The materials and methods section should include the total number of patients, inclusion and exclusion criteria and details about the selected patients (some information is mistakenly placed under the results section).

4.       The statistical analysis is too basic, with few statistically significant findings and no clear correlations. Add logistic regressions, ROC curves, and other advanced analyses to justify the results.

5.       The number of patients is too small for such a topic.

6.       Figures 2 and 3 should be removed as they are not relevant to this article.

7.       The discussion section could include more valuable information from the literature, particularly from studies with larger patient cohorts.

8.       The conclusions are not supported by the overly simplistic results and are easy to challenge.

9.       The title should not contain a comma before "and."

Author Response

Dear Reviewer,

We appreciate the time and effort that you dedicated to providing feedback on our manuscript and are grateful for the insightful comments and valuable improvements to our paper. We have made the relevant changes according to your suggestions. Those changes are highlighted with track changes function through the manuscript. Please see below for a point-by-point response to the comments and suggestions.

Reviewer report:

Comments and Suggestions for Authors

The article "Relationship between sigmoid volvulus subtypes, clinical course, and imaging findings" provides information about sigmoid volvulus, along with clinical and imaging aspects of this pathology. Recommendations:

Comment: The article has a chaotic structure and needs reorganization. Decide whether to focus on a statistical analysis of relationships between elements or on case presentations.

Response: In accordance with your and other reviewers' recommendations, the manuscript has been restructured to emphasise the original research.

Comment: The introduction is overly long, uses simplistic language and contains many details that do not provide real value to the reader.

Response: The introduction has been streamlined by eliminating superfluous details and enhancing the article's fluency.

Comment: The materials and methods section should include the total number of patients, inclusion and exclusion criteria and details about the selected patients (some information is mistakenly placed under the results section).

Response: Some of the data that were mistakenly included in the results section have been brought to your attention, and we would like to extend our sincere apologies for this oversight. All of this information, including the total number of patients, has been inserted into the materials and methods section. We have incorporated the inclusion and exclusion criteria, along with more comprehensive details regarding the selected patients, in the materials and methods section as follows.

“Our study will include patients with surgically confirmed sigmoid volvulus who un-derwent a contrast-enhanced abdominal CT examination at our hospital within 24 hours of symptom onset, and whose follow-up and treatment occurred at our facility, between January 2016 and January 2024. Individuals whose clinical and radiologic findings indicated the presence of SV but not a volvulus was observed during the ope-ration, as well as individuals whose necessary records could not be accessed due to their ongoing treatment in another healthcare facility subsequently excluded from the study”

Comment: The statistical analysis is too basic, with few statistically significant findings and no clear correlations. Add logistic regressions, ROC curves, and other advanced analyses to justify the results.

Response: We appreciate your significant observations regarding the simplicity of our statistical analyses and the absence of distinct correlations. We intended to incorporate additional advanced analyses, such as logistic regressions and ROC curves, to substantiate the results; however, the patient sample size in the study was adequate solely for descriptive and preliminary numerical analysis. These analyses were not preferred for this reason. Furthermore, the limited patient population, particularly within the organoaxial group, was noted as a limitation.

Comment: The number of patients is too small for such a topic.

Response: Your apprehension regarding the necessity for a larger patient cohort for a subject requiring evaluation within such an extensive framework is entirely justified. Due to the relatively rigorous inclusion criteria, the number of patients who were able to participate in this study was reduced. That resulted in a decrease in the statistical power of the study when compared to other studies in the literature that were comparable and had larger cohorts, particularly for the organoaxial sigmoid volvulus. This is a limitation that has been added.

Comment: The discussion section could include more valuable information from the literature, particularly from studies with larger patient cohorts.

Response: In accordance with your recommendations, valuable information from studies involving larger patient cohorts has been incorporated into the discussion section as follows.

“The most recent research conducted by Moloney et al. on mesentero-axial and organo-axial SV involving 117 patients evaluated the direction of rotation in the axial and coronal planes. The authors have demonstrated a significant correlation between clockwise rotation in one plane and anticlockwise rotation in the opposing plane. They also emphasised that these rotational directions are significant findings in identifying the type of volvulus. It has been shown that the mesentero-axial SV subtype is linked to anti-clockwise rotation in the axial plane and clockwise rotation in the coronal plane, while the organo-axial SV subtype is associated with clockwise rotation in the axial plane and anti-clockwise rotation in the coronal plane. The determination of these rotation directions, first established in the literature, may serve as a reliable alternative to the number of transition zones, which is the most definitive indicator in distinguis-hing organo-axial and mesentero-axial SV subtypes, as demonstrated in our study.”

“The study by Atamanalp et al., conducted in our geographical region and representing the largest single-centre experience globally with over 1000 SV patients, demonstrated that the treatment success rates of non-operative and operative detorsions were com-parable in patients without intestinal ischaemia. Atamanalp et al. and Moloney et al. demonstrated that endoscopic detorsion techniques yielded lower mortality and morbidity rates. At this point, it is crucial to rule out ischaemia on CT scans to facilitate the referral of the patient for endoscopic detorsion. Conversely, the presence of ischaemic findings necessitates surgical inter-vention, which should be conducted promptly to mitigate mortality risks.”

Comment: The conclusions are not supported by the overly simplistic results and are easy to challenge.

Response: Conclusion section has been changed as follows to be supported with results.

“In summary, our research indicates that the mesentero-axial variant of sigmoid volvulus is more prevalent than the organo-axial variant in our region, which is re-cognised as endemic for sigmoid volvulus. According to our results, CT can be a usefull tool to differentiate organoaxial and mesenteroaxcial subtypes.”

Comment: The title should not contain a comma before "and."

Response: We apologise for this simple mistake. The necessary correction has been made.

Round 2

Reviewer 3 Report

Comments and Suggestions for Authors

The revised version of the article has improvements over the previous model, but still the quality of the material remains quite low and many of the comments were not properly addressed.